REGISTERED REPORT PROTOCOL

# Situation analysis for delivering integrated comprehensive sexual and reproductive health services for displaced population of Kasaï, Democratic Republic of Congo: Protocol for a mixed method study

Jacques B. O. Emina[1,2]*, Parfait Gahungu[3], Francis Iyese[1], Rinelle Etinkum[1], Brigitte Kini[4], Joseph Fataki Mobolua[5], Mohira Boboeva[6], Loulou Kobeissi[7]

**1** Population and Health Research Institute, Kinshasa, Democratic Republic of Congo (DRC), **2** Department of Population and Development Studies, University of Kinshasa, Kinshasa, Democratic Republic of Congo (DRC), **3** Programme National de Santé des Adolescents (PNSA), Kinshasa, Democratic Republic of Congo (DRC), **4** WHO National Office, Kinshasa, Democratic Republic of Congo (DRC), **5** Projet SR en situation d'urgence dans la Région du Kasaï, WHO- DRC, Kinshasa, Democratic Republic of Congo (DRC), **6** Global Health Cluster, Emergency Operations Department, Health Emergencies Program, World Health Organization (HQ), Geneva, Switzerland, **7** Department of Sexual and Reproductive Health and Research (SRH), including the UNDP/UNFPA/UNICEF/WHO/World/Bank Special program of research, development and research training in human reproduction (HRP), World Health Organization (HQ), Geneva, Switzerland

* Jacques.emina@gmail.com

This is a Registered Report and may have an associated publication; please check the article page on the journal site for any related articles.

## Abstract

## Introduction

Delivering integrated sexual and reproductive health services (SRHS) in emergencies is important in order to save lives of the most vulnerable as well as to combat poverty, reduce inequities and social injustice. More than 60% of preventable maternal deaths occur in conflict areas and especially among the internally displaced persons (IDP). Between 2016 and 2018, unprecedented violence erupted in the Kasaï's region, in the Democratic Republic of Congo (DRC), called the Kamuina Nsapu Insurgency. During that period, an estimated three million of adolescent girls and women were forced to flee; and have faced growing threat to their health, safety, security, and well-being including significant sexual and reproductive health challenges. Between August 2016 and May 2017, the "Sous-Cluster sur les violences basées sur le genre (SC-VBG)" in DRC (2017) reported 1,429 Gender Based Violence (GBV) incidents in the 49 service delivery points in the provinces of Kasaï, Kasaï Central and Kasaï Oriental. Rape cases represented 79% of reported incidents whereas sexual assault and forced marriage accounted for respectively 11% and 4% of Gender Based Violence (GBV) among women and adolescent girls. This study aims to assess the availability of SRHS in the displaced camps in Kasaï; to evaluate the SRHS needs of young girls and women in the reproductive age (12–49). Studies of sexual and reproductive health (SRH) in the Democratic Republic of Congo (DRC) have often included adolescent girls under the age of 15 because of high prevalence of child marriage and early onset of childbearing,

**Data Availability Statement:** All relevant data from this study will be made available upon study completion. The approved report and datasets will be posted on the WHO website (www.who.int) and Population and Health Research Institute (PHERI) website (www.pheri.org).

**Funding:** This work was funded by the Department of Reproductive Health and Research (RHR), including the UNDP/UNFPA/UNICEF/WHO/World/ Bank Special programme of research, development and research training in human reproduction (HRP). The funders had no role in study design, data collection and analysis, decision to publish, or preparation of the manuscript.

**Competing interests:** The authors have declared that no competing interests exist. The authors alone are responsible for the views expressed in this article and they do not necessarily represent the views, decisions or policies of the funding bodies or institutions with which they are affiliated.

especially in the humanitarian context. According to the 2013 Demographic and Health Survey (DHS), about 16% of surveyed women got married by age 14 while the prevalence of early child marriage (marriage by 15) was estimated at 30%; to assess the use of SRHS services and identify barriers as well as challenges for SRH service delivery and use. Findings from this study will help provide evidence to inform towards more needs-based and responsive SRH service delivery. This is hoped for ultimately improve the quality and effectiveness of services, when considering service delivery and response in humanitarian settings.

## Data and methods

We will conduct a mixed-methods study design, which will combine quantitative and qualitative approaches. Based on the estimation of the sample size, quantitative data will be drawn from the community-based survey (500 women of reproductive age per site) and health facility assessments will include assessments of 45 health facilities and 135 health providers' interviews. Qualitative data will comprise materials from 30 Key Informant Interviews (KII) and 24 Focus Group Discussions (FGDs), which are believed to achieve the needed saturation levels. Data analysis will include thematic and content analysis for the KIIs and FGDs using ATLAS.ti software for the qualitative arm. For the quantitative arm, data analysis will combine frequency and bivariate chi-square analysis, coupled with multi-level regression models, using Stata 15 software. Statistic differences will be established at the significance level of 0.05. We submitted this protocol to the national ethical committee of the ministry of health in September 2019 and it was approved in January 2020. It needs further approval from the Scientific Oversee Committee (SOC) and the Provincial Ministry of Health. Prior to data collection, informed consents will be obtained from all respondents.

## Introduction

Delivering integrated sexual and reproductive health services (SRHS) in emergencies saves lives of the most vulnerable, reduce inequities as well as helps to combat poverty and social injustice. It is estimated that more than 60% of preventable maternal deaths occur in conflict areas and among the displaced people [1, 2].

Between 2016 and 2018, unprecedented levels of violence erupted in the Kasaï region, in the Democratic Republic of Congo (DRC), commonly known as the Kamuina Nsapu Insurgency. Medias, Civil Society Organizations (CSO) and Non-Governmental Organizations (NGOs) reported massacres, public executions, and rapes. During this period, about three million adolescent girls and women were forced to be displaced; and faced growing threat to their health, safety, security, and well-being including significant sexual and reproductive health challenges [3]. Between August 2016 and May 2017, the "Sous-Cluster sur les violences basées sur le genre [3] reported 1,429 Gender Based Violence (GBV) incidents in the 49 service delivery points in the provinces of Kasaï, Kasaï Central and Kasaï Oriental. Rape cases represented 79% of reported incidents, whereas sexual assault and forced marriage represented respectively 11% and 4% of Gender Based Violence (GBV) among women and adolescent girls.

With this context in mind, the world health organization (WHO) and its partners through the Health Cluster, are working in three countries (Cox-Bazar in Bangladesh, Yemen, and the region of Kasaï in the Democratic Republic of Congo) to strengthen its internal capacity and that of local health providers to enhance the provision and delivery of SRHR services and to

help reduce unmet needs among IDP [1]. In the Kasaï region of the DRC, the 2013 Demographic and Health Survey (DHS) conducted before the Kamuina Nsapu's insurgency reported poor reproductive health indicators [4]. The region had the highest total fertility rate (8 children per women versus the national average of 6.6 children per woman) and significantly low modern contraceptive prevalence (5% against the national average of 8%).

An SRH needs assessment exploring access to and availability of services, gaps, challenges and services outcomes is essential among displaced adolescent girls and women living in the Kasaï's region (Kasaï, Kasaï Central and Kasaï Oriental) in DRC. This assessment will provide an overview of the status of SRH needs among displaced young girls and women, and highlight the extent of available SRH services in Kasaï, Kasaï Central and Kasaï Oriental.

The specific objectives of this assessment are to: a) assess the availability of SRH services in areas where displaced people are living in the three provinces; b) evaluate the SRH needs of displaced girls and women of reproductive age (12–49); c) identify barriers and challenges in the field of SRH in villages with concentration of displaced people; and d) describe current provider's capacity to provide SRH services (including health workers capacity and other issues as specified as well) in those villages.

## Methodology

### Study design

This research will consist of a mixed-methods study design, combining quantitative and qualitative approaches. Quantitative data will comprise of community-based surveys among young girls and women and health facility assessments. Qualitative data will comprise of Focus Group Discussions (FGDs) and key informant interviews (KIIs) with a broad range of stakeholders involved in SRH service delivery and response.

The Clinic-Based Family Planning and Reproductive Health Services in Africa models will guide the design of the related data collection tools. Fig 1 shows the study conceptual framework adapted from "Clinic-Based Family Planning and Reproductive Health Services in Africa" [5].

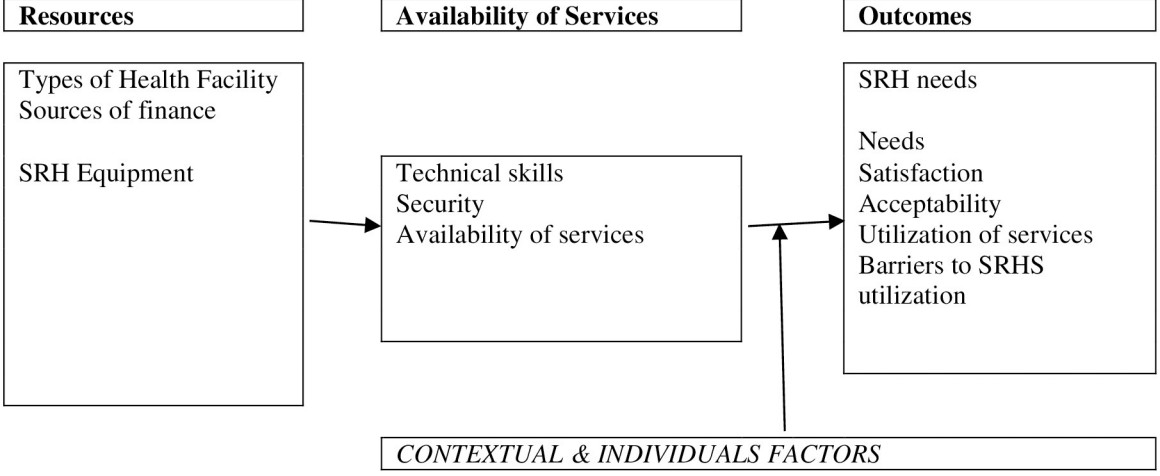

**Fig 1.**

Each component of the conceptual framework is in line with the study objectives. Resource variables encompass the program inputs and processes, such as those related to human resources and infrastructure.

SRH services variables will explore antenatal care, postnatal care, post-abortion care, Family Planning, Sexually transmitted infection (STI) management, etc. The main study outcome variables include the following: SRH health Knowledge, SRH service seeking behavior and barriers to use SRHS among displaced women in Kasaï.

## Study site

The study will be conducted in displaced camps/ villages in the region of Kasaï, including the Kasaï, Kasaï Central and Kasaï Oriental provinces.

The province of Kasaï has an area of 95,631 km$^2$ and has an estimated population of four million, of which approximately 40% are displaced. It includes five territories: Dekese, Ilebo, Luebo, Kamonia and Mweka. It has 18 health zones (HZ), among which 11 were affected by the Kamwina Nsapu conflict [6, 7]. The city of Tshikapa is its capital town.

Kasaï Central's province measures 59,111 km$^2$ and has an estimated population of four million inhabitants. This province is constituted of the following five territories: Tshimbulu, Demba, Dibaya, Kazumba and Luiza. The Kasaï Central's province has 26 HZs, of which 24 have been affected by the conflict [6, 7]. Kananga is the city capital. In Kasaï Central, the survey will cover essentially health zones around Kananga, the city capital.

Kasaï Oriental's province has an area of 9,545 km$^2$ occupied by about five million inhabitants. Five territories are part of it; listed as Tshilenge, Lupata pata, Katanda, Miabi and Kabeya Kamwanga [6, 7]. The last two territories were the most affected by the conflict. It has 19 HZs, eight of which were affected by the conflict, i.e. 50%. The city of Mbuji Mayi is its chief town.

## Participants—Target population

The study will sample different target population across both the quantitative and qualitative assessments with the aim of triangulation of obtained findings. The main study population includes adolescent girls and women of reproductive age 12–49 living in displaced camps in the province of Kasaï, Kasaï Central, and Kasaï Oriental provinces. Though studies of sexual and reproductive health, including the DHS usually focus on women aged 15–49, this study will include adolescent girls and women aged 12–49. This is particularly triggered as a result of the observed high prevalence of child marriage and early onset of childbearing rates in the Democratic Republic of Congo (DRC), as a result of the prevailing humanitarian context in the selected three the Kasaï's region of this assessment. According to the 2013 Demographic and Health Survey (DHS), the overall prevalence rates of early child marriage, defined as marriage before the age of 15, among the surveyed women were 16%; while 30% of women were married before the age of 18 [4]. This population (12–49) will be eligible to participate in the community-based survey and in FGDs.

Further, data collection will also target health providers, leaders of opinion, such as community and religious leaders, and married men living in the same displaced zones. The Health providers will be selected to participate in the health facility assessments. Community leaders will be selected to participate in the KIIs and married men will be selected to participate in the FGDs.

## Data sources

The study will combine multiple data sources. We will use both the quantitative and qualitative data that will serve as primary sources. The secondary data sources will be the quantitative

data from health facilities, NGO and other institutions' (WHO, UNFPA, UNICEF, etc.) reports as well as documents from the ministry of social affairs and the ministry of Public Health in DRC. Information from those sources will help comprehend and describe the study context, status of health services and their use in general, and among women in particular, as the compiled SRH services use information will be area specific.

As already indicated, the quantitative data will come from both the community surveys and the health facility assessments. The qualitative data will be collected using Key Informant Interviews (KII) and Focus Group Discussions (FGDs).

**Quantitative surveys.** *Community-based survey*. We will carry out a community-based survey among adolescent girls and women in the Kasaï region will be carried out. The sampling strategy and sampling size calculation are outlined, as per the below formula.

This formula determines the minimum sample size needed for community-based quantitative survey:

$$n = \frac{z^2 * pq}{e^2} * DE$$

- z (z-score) is the number of standard deviation (error) by which an estimate is above or below the average, assuming normal distribution. Z has the standard normal distribution with mean zero and variance one, $z \sim N(0,1)$. Given the significance level, α, z-score depends on whether it is a one-sided, $z = F^{-1}(1 - \alpha)$, or two-sided test (effect), $z = F^{-1}(1 - \frac{\alpha}{2})$, where $F(z)$ is the cumulative distribution function of z. For this study, the significance level, α, is set at 0.05, where z-score for two-sided test is 1.96 and for one-sided test z is 1.645;

- e is the margin of error calculated by multiplying the z-score and the standard error of the estimate. In the above formula, e is fixed based on the level of precision we would like to achieve;

- p is the proportion of women and adolescents (12–49) with SRH needs. In absence of accurate data, this proportion is estimated set at 0.50 (50%);

- q is 1 –p, proportion of a women and adolescents (12–49) without SRH needs.

- DE is the design effect. The design effect is the ratio of the variance of the estimate under a specific (given) design over the variance of the estimate under the simple random sampling.

Assuming the confidence level of 95% (Z = 1.96), the marginal error of 5% (0.05), the refusal rate of about 10% and the design effect of 1.2, the minimum estimated sample size needed for this survey is estimated to 480 young girls and women of reproductive age (12–49) per site. We rounded up the number to 500 women.

We will select 500 households randomly distributed across the selected villages where displaced people are living. We will use a multi-stage cluster sample. The sampling will revolve a random selection of villages followed by random selection of households by random from each province. First, we will select randomly 20 villages from an updated list of villages/camp where displaced population are living) in each province. Then, we randomly choose 25 households in each selected village/camp. In total, 500 households will be selected (20 villages X 25 households) per province. To be able to randomly select households in a village, a list of all households and their location (structure number) will be prepared prior to actual fieldwork (mapping and numbering). Finally, following the households' selection, within each selected household, we will interview all women aged 12–49.

The community-based survey hast two questionnaires: the household questionnaire (S1 Annex, S1 Appendix) and the woman questionnaire (S2 Annex, S1 Appendix).

The household questionnaire contains a cover sheet to help identify the household and it lists all members of the household and visitors (Household Schedule). It seeks to collect socio-demographic and housing characteristics information. The goal of the questionnaires is to collect not only socio-demographic information such as name, sex, age, education, marital status and relationship to the head of household, but also information related to the housing characteristics. The housing characteristics information of interest are type of water source, sanitation facilities, quality of flooring, and ownership of durable goods. The household questionnaire (S1 Annex, S1 Appendix) will be administered to the head of household or any other credible respondent. In addition, it will also assist the interviewer to identify young girls and women respondents who will be eligible for the Woman Questionnaire. Adolescent girls and women aged 12–49 years in every selected household who are members of that household (i.e. those that usually live in the household) and visitors (those who do not usually live in the household but who slept there in the previous night) will be eligible for interview.

As indicated, the women questionnaire (S2 Annex, S1 Appendix) will be separately administered to all females in the household who are of reproductive age (12–49 years old). The Woman's Questionnaire seeks to collect data on the following topics:

- Socio-demographic characteristics: age, marital status, education, employment, media exposure, place of residence and migration information.

- Reproductive behavior and intentions: Questions will cover dates of all still and live births, abortions and miscarriages over the last three years (2016–2019), current pregnancy status, fertility preferences, and future childbearing intentions of each woman of reproductive age (12–49 years).

- Utilization of SRH Services: questions will cover antenatal and postnatal care, place of delivery, person who attended the delivery, birth weight, and the nature of complications during pregnancy for recent births.

- Family planning: questions will cover knowledge and use of specific contraceptive methods, source of contraceptive methods, exposure to family planning messages, informed choice, and unmet need for family planning. For women not using contraception, we include questions on knowledge of a source of contraception and reasons for no use.

- Menstrual regulation and abortion section: questions will cover knowledge and practices of menstrual regulation as well as abortion, methods used and consequences.

- Sexually transmitted infections (STIs): questions will cover knowledge, experience of STIs and health seeking behavior.

- Adolescent sexual and reproductive health: questions will cover menstrual health, exposure to comprehensive sexuality education and attitudes toward SRH.

*Health facility surveys.* The health facility assessment will combine a health facility quantitative survey coupled with a facility audit.

A total of 15 health facilities per province will be selected. These will include one referral hospital and health centers. In each selected health facility, an audit and a providers' interview will be conducted.

**Facility audit/inventory.** The Facility audit (inventory) questionnaire (S3 Annex, S1 Appendix) will collect general information on the health facility, staffing, availability of SRHS, safety and hygiene, diagnosis of sexual transmitted infections and human immunodeficiency

virus (STIs/ HIV), equipment and supplies, the use of SRHS as well as indicators for measuring the quality of care and stock-outs. The facility audit questionnaire will combine both questions for the facility manager and it will be coupled and the standardized assessment of the health facility. Only health facilities located in villages where displaced people are living will be selected.

**Providers' interview.** We will interview a total sample of **60 providers in each site** (4 providers X 15 health facilities) will be interviewed. In each site, all providers will be listed and assigned a number. Using Stata 15 command for random sample selection, four providers per facility will be selected to complete the questionnaire. If the number of health providers is less than four in a health facility, all providers will be eligible for the survey. Providers are eligible for selection, if they manage one or more of the following SRH services: working in family planning services, outpatient clinic; post-partum; post-abortion; and other health areas if services are integrated. The data collected from the providers will include: information on their characteristics and qualifications; perceived views of the program and confidence in training received; experience and attitudes toward the service delivery environment; perceived challenges; and future recommendations.

In particular, the questionnaire will be divided into the following sections: Providers' background characteristics, including education and experience; General pre-service and in-service training; experience in Family planning services; experience in maternal health; experience in sexuality transmitted infections, including HIV and acquired immunodeficiency syndrome (AIDS); Diagnosis services and working conditions (See S4 Annex for the providers' question, S1 Appendix).

**Qualitative surveys.** The Qualitative surveys will consist of 30 Key Informant Interviews (KIIs) and 12 Focus Group Discussions (FGDs). Box 1 summarizes the content of each group and the respective sampling process.

The KIIs will be conducted as semi-structured interviews with principal stakeholders, including religious leaders (priests and pastors), school managers and teachers, health professional, journalists, staff of NGOs working on sexual and reproductive health, religious leaders, youth leaders, civil society organizations, lay public males and females. They will also be conducted among key actors and stakeholders from different NGOs, for example, staff of

### Box 1. Sampling frame and characteristics for qualitative activities

| Activity | Sampling Strategy | Respondent Groups | Estimated Numbers/ site |
|---|---|---|---|
| Key Informant Interviews (KIIs) | Opportunistic/ purposive emergent sampling | Influential workers, SRH and GBV group, UNFPA & WHO focal points | 10 people per province |
| | | community: religious leaders, teachers, Health Professional (formal and informal, CHWs) | |
| Focus Group Discussions (FGDs): *to understand SRHR services, providers (external and internal), violence, young men's role in camp (frustration, stressors), do they use services, cost of services, general expectations from work & life)* | Purposive sampling | Males (less than 25; 25–59), Females (Less than 25; 25–49). | 8 FGDs per province |

international organizations (such as SRH Working Group (WG) led by UNFPA, WHO, etc.) to provide an insight regarding the service availability and utilization.

A total of 30 key informants will be selected (as indicated), 10 per province/ site. The selection of the key informants will be based on the following criteria: (1) A member of the local community; (2) knowledge and involvement in sexual and reproductive health related activities; (3) a professional or leader of opinion (member of a local or international organization, teacher, community leader, churchmen, etc.); and (4) provided consent for the interview.

FGDs will also be conducted with displaced married men and women. These FGDs aim to provide an overview about men and women prevailing perceptions around access to SRH services, needs and use among the families. Furthermore, they will capture the level of male engagement in promoting sexual and reproductive health and rights; understand men's views of their preferred methods of engagement; as well as explore factors that could enable better men's engagement in Family Planning (FP) decision-making, prevention of Gender-based violence, and prevention and management of STI/HIV/AIDs. Questions exploring issues around the importance of sexual and reproductive health programs, particularly in humanitarian crisis context (perceived benefits, perceived threats of SRH problems, perceived barriers) regarding implementation of SRH and mitigation strategies, and sustainability aspects of SRH programs/ interventions will also be discussed during the FGDs. The FGD guide is found in S1 Appendix.

In total, we will conduct 24 focus group discussions (FGDs), eight per province based on age and sex of participants. Four FGDs will be conducted with men and four with women. For each age group, four FGDs will be conducted with adults and four with youth. Two FGDs for young females, two FGDs for young men, two FGDs for men (25 year of age or above), two FGDs for women (25 year of age or above). This is consistent with Hennink (2019) [8] who suggested that two groups per stratum provided a more comprehensive understanding of issues, while more groups per stratum provided little additional benefit, particularly in mixed studies. However, we will conduct further FGD until saturation of information, if saturation is not reached after 2 FGD per age group. The selection of the FGDs participants will be based on age, gender and the fact of being displaced because of the Kamwina Nsapu's conflict.

The FGDs will be conducted in Tshiluba, the national language spoken in the three Kasaï's region which comprises the three provinces of Kasaï, Kasaï Central and Kasaï Oriental. The collected materials will further be translated and transcribed from Tshiluba to French using Microsoft Word.

## Data analysis methods

Data analysis combine both qualitative and quantitative statistical analyses. Qualitative data analysis will include thematic and content analysis of KIIs and FGDs using ATLAS.ti software. Quantitative data analysis will combine statistical techniques of mainly univariate and bivariate analysis, including t-test, chi-square test, survival analysis, multi-level regression models, analysis of variance (ANOVA) and Tree Decision analysis depending on the nature of the dependent variable and the specific. The level of significance is set to 0.05. We will use Stata 15 (StataCorp, 4905 Lakeway Drive, College Station, Texas, USA) for data cleaning, management and analysis.

## Ethical considerations

The research protocol and data collection tools have been submitted to the National Ethical Committee of the Ministry of Health in DRC in September 2019. Prior to data collection, a formal authorization request will be sought from the central and local administrations of the Ministries of Public Health (MoPH), Interior Affairs (MIA) and Ministry of Social Affairs (MSA).

Furthermore, the Scientific Oversee Committee (SOC), including experts from the ministry of health, International Non-Governmental Organizations (NGOs) working on Sexual and Reproductive Health (SRH) and in humanitarian settings as well as experts from Universities will provide support, guidance and oversight progress. Prior to data collection, informed verbal consent will be obtained from all respondents coupled with ensuring maximum levels of privacy and confidentiality of the different study participants.

At the field level, informed verbal consent will be obtained from all respondents, whether the community-based (household') survey, the KIIs and the FGDs. Written informed consent will be also asked for health facility surveys. All participants will be ensured that they have the full right to participate or decline participation without losing any benefits or rights. Individuals choosing to participate will be informed that they are free to withdraw from the study at any point of time, and they can choose to not answer questions if they feel uncomfortable. Subjects will NOT be asked to give permission for release of identifiable data at any time during the study duration or in the future. The individual's privacy will be strictly respected during data collection. Data collectors will be properly trained to respect participants' privacy and right to immediately end the study if privacy is violated. Confidentiality will be maintained at each step of data collection and data analysis. We will use appropriate disclosure avoidance methods to prevent any linkage between the data and individual respondents. There will not be any identifiers to link data with individuals. All data will be under lock and key's access will be given only to the research team.

## Data quality

In order to ensure data quality, the research team will employ various quality control measures throughout the study process. The following strategies will be used to monitor and preserve data quality standards: (1) pre-testing of study instruments to ensure relevance and validity; (2) electronic data collection tools (with tablets) using software and computer validation programs to check the logical consistency of data; (3) training and assessment of fieldworkers prior to actual data collection; (4) close supervision of fieldworkers (One supervisor will oversee five data collectors); (5) organization of daily debriefing meeting for all field staff to share experiences, lessons learned and challenges; (6) accuracy, consistency and completion will be conducted for both quantitative and qualitative data at the end of each day; (7) analysis of the completeness of information as well as identification of outliers during data cleaning and data analysis. Qualitative data will be transcribed and translated into French and English prior to analysis.

## Expected challenges and mitigation strategies

Three main challenges could be anticipated during the implementation of this research, which could directly impact data collection. Those challenges are the ability to access and recruit the desired displaced population, language, and accessibility of the data collection sites.

- First, the location and the ability to identify and recruit the research target populations. In the three provinces, some displaced people are lodged in family dwellings, which may not be easily located although the large majority are living in well located areas in villages. Thus, we will coordinate and seek the assistance of the local leaders to identify villages and households where the displaced people are living in Kasaï, Kasaï Central and Kasaï Oriental province. Collaboration with humanitarian organizations working with those displaced population will also be sought to facilitate identification of these sites as well as of displaced people.

- The second challenge pertains to language. Although this protocol and the final report will be written in English, the research proposal and respective data collection tools that are needed for submission to the administrative authorities will all be translated into French. Moreover, in the field, all data collection will be conducted in Tshiluba. We will hire supervisors and fieldworkers who are able to speak both Tshiluba and French. We will also, pay for translation fees and/or for specialists to ensure appropriate and accurate translation.

- Finally, the third challenge concerns the accessibility to the sites due to the poor road infrastructure and the poor anticipated weather conditions, the start of the raining season. We will do our best to complete the data collection before the deterioration of the weather conditions.

## Strengths and limitations of this study

- This situation analysis is among the first to provide an overview of adolescent girls and women's (aged 12–59 years old) SRH needs and issues, availability and delivery of SRH services, barriers to service uptake and related challenges in the three provinces of Kasaï in the Democratic Republic of Congo, which is in humanitarian crisis situation due to people's displacement.

- The study will use a concurrent mixed-methods study design to assess the current situation and understand the community perspectives and facility readiness to provide different SRHS and identify related gaps and challenges.

- A potential limitation can be foreseen during this assessment could pertain to the unwillingness of certain respondents, both at the community level and/or facility levels, to disclose sensitive information related to SRH practices, service utilization and health facility records.

## Supporting information

**S1 Appendix.**
(DOCX)

**S1 Annex.**
(ZIP)

**S2 Annex.**
(DOCX)

**S3 Annex.**
(DOCX)

**S4 Annex.**
(PDF)

## Acknowledgments

In memory of Parfait Gahungu Ndongo passed away on May 21, 2020. We are grateful to Prof. Blandine Bawawana Bavwindinsi and Dr. Didier Mbayi Kangudie for reviewing this manuscript.

## Author Contributions

**Conceptualization:** Jacques B. O. Emina, Parfait Gahungu, Francis Iyese, Rinelle Etinkum, Joseph Fataki Mobolua, Mohira Boboeva, Loulou Kobeissi.

**Funding acquisition:** Jacques B. O. Emina.

**Methodology:** Jacques B. O. Emina, Parfait Gahungu, Francis Iyese, Rinelle Etinkum, Brigitte Kini, Joseph Fataki Mobolua, Mohira Boboeva, Loulou Kobeissi.

**Resources:** Brigitte Kini, Mohira Boboeva.

**Supervision:** Jacques B. O. Emina.

**Validation:** Jacques B. O. Emina, Mohira Boboeva, Loulou Kobeissi.

**Writing – original draft:** Jacques B. O. Emina, Loulou Kobeissi.

**Writing – review & editing:** Jacques B. O. Emina, Parfait Gahungu, Francis Iyese, Rinelle Etinkum, Brigitte Kini, Joseph Fataki Mobolua, Mohira Boboeva, Loulou Kobeissi.

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
