## [Decision Letter · Decision Letter 0]

24 Jul 2020

PONE-D-20-10507

Situation Analysis for Delivering Integrated Comprehensive Sexual and Reproductive Health Services for displaced population of Kasaï, Democratic Republic of Congo: Protocol for a mixed method study

PLOS ONE

Dear Dr. Emina,

Thank you for submitting your manuscript to PLOS ONE. After careful consideration, we feel that it has merit but does not fully meet PLOS ONE’s publication criteria as it currently stands. Therefore, we invite you to submit a revised version of the manuscript that addresses the points raised during the review process.

We look forward to receiving your revised manuscript.

Kind regards,

Philip Anglewicz, PhD

Academic Editor

PLOS ONE

Journal Requirements:

2. At this time, we require the following information in order to proceed with your submission: please confirm that you have received the necessary ethics approval mentioned in your ethics statement. In addition, please seek ethics approval from your own institution's IRB; namely, please ensure that your project is reviewed and approved by the University of Kinshasa's IRB. Thank you for your attention to this request.

3.In your Data Availability statement, you have not specified where the minimal data set underlying the results described in your manuscript can be found. PLOS defines a study's minimal data set as the underlying data used to reach the conclusions drawn in the manuscript and any additional data required to replicate the reported study findings in their entirety. All PLOS journals require that the minimal data set be made fully available. For more information about our data policy, please see http://journals.plos.org/plosone/s/data-availability.

5. Your ethics statement must appear in the Methods section of your manuscript. If your ethics statement is written in any section besides the Methods, please move it to the Methods section and delete it from any other section. Please also ensure that your ethics statement is included in your manuscript, as the ethics section of your online submission will not be published alongside your manuscript.

6.We note that [Figure(s) 2] in your submission contain [map/satellite] images which may be copyrighted. All PLOS content is published under the Creative Commons Attribution License (CC BY 4.0), which means that the manuscript, images, and Supporting Information files will be freely available online, and any third party is permitted to access, download, copy, distribute, and use these materials in any way, even commercially, with proper attribution. For these reasons, we cannot publish previously copyrighted maps or satellite images created using proprietary data, such as Google software (Google Maps, Street View, and Earth). For more information, see our copyright guidelines: http://journals.plos.org/plosone/s/licenses-and-copyright.

1.    You may seek permission from the original copyright holder of Figure(s) [2] to publish the content specifically under the CC BY 4.0 license. 

Reviewers' comments:

Reviewer's Responses to Questions

**Comments to the Author**

1. Does the manuscript provide a valid rationale for the proposed study, with clearly identified and justified research questions?

Reviewer #1: Yes

2. Is the protocol technically sound and planned in a manner that will lead to a meaningful outcome and allow testing the stated hypotheses?

Reviewer #1: Yes

3. Is the methodology feasible and described in sufficient detail to allow the work to be replicable?

Reviewer #1: No

4. Have the authors described where all data underlying the findings will be made available when the study is complete?

Reviewer #1: No

5. Is the manuscript presented in an intelligible fashion and written in standard English?

Reviewer #1: No

6. Review Comments to the Author

You may also provide optional suggestions and comments to authors that they might find helpful in planning their study.

Reviewer #1: Dear authors,

The protocol for the situation analysis of SRHS for displaces population of Kasaï is an interesting topic that deserves full assessment to improve such health care delivery for displaced women. From the description this project is very ambitious and a huge amount of work will be required to pull this off, I’m assuming that the authors have sufficient time, human and financial resources to fulfil all the work.

The manuscript needs substantial revision for the English language, the manuscript should be edited by an English native speaker.

My concern in general is that this situation analysis concerns SHR health care delivery. However, from the protocol it was not very clear if the situation analysis of health care facilities that will be visited, if they were specifically set-up in these camps to deliver health care for the displaces population? Or if staff has received specific training for the care of post-conflict, displaced individuals? From reading the manuscript, the feeling is that it is a general SHR assessment of SHR rather than SHR assessment for displaced people. It seems that the health care facilities that will be visited and audited are the ones that are traditionally there to provide healthcare to the resident population. Then the outcome could be biased because of course these type of healthcare facilities will not be prepared to deliver post-traumatic healthcare.

Another concern is the focus of the study, data on SHR delivery will be collected from a large variety of sources, which is an advantage for triangulation, however the questionnaires of Qualitative research are very exhaustive collecting insights on SHR delivery for men, women, adolescent boys and girls. From the introduction I understood that the target study population was displaced women, while the Qualitative study is trying to gather insight on SHR delivery in general, this related back to my first concern. Do you want to know about SHR delivery in general? Or specifically for the displaced female population?

It is not explained in the text why you are sampling women and young girls from 12-49 years of age for the women questionnaire while the WHO definition of women of reproductive age is 15-49 years of age.

Lastly, my major concern is about the sampling of households and women for the community based survey. In the methods you describe that you will select from an updated list of villages/camps. Already in the introduction it is not well described if the displaced are living in villages (permanent housing) or in camps (temporary housing). The description of the context with a bit more detail will help to understand the methodology used. Later on in the expected challenges section you describe that the location and the ability to identify and recruit the target populations could be a challenge. The mitigation plan is to involve leaders and NGO’s, but my concern is how will you be able to apply the random sampling strategy if this housing list, village list, camp list are not available?

More specific comments will be uploaded in a separate file.

7. PLOS authors have the option to publish the peer review history of their article (what does this mean?). If published, this will include your full peer review and any attached files.

Reviewer #1: No

---

## [Author Response · Author response to Decision Letter 0]

23 Sep 2020

1. At this time, we require the following information in order to proceed with your submission: please confirm that you have received the necessary ethics approval mentioned in your ethics statement. In addition, please seek ethics approval from your own institution's IRB; namely, please ensure that your project is reviewed and approved by the University of Kinshasa's IRB. Thank you for your attention to this request.

Answer: Thank you for your comment. We have received the ethical approval from the national ethical committee of the Ministry of Health established by Ministerial order n ° 1250 / CAB / MIN / S / ZKM / 043 / MC / 2006 of December 18, 2006. It is registered under (HHS: IORG00088558 / IRB) at the US Department of Health and Human registration number. 

Furthermore, we have received authorization to conduct the study signed by the Secretary General of the Ministry of Health. We detail this in the manuscript.

This study will be conducted by Population and health Research Institute (PHERI), an independent research firm based in Kinshasa, Democratic Republic of Congo and The National Adolescent Health Programme /Programme National de Santé de l’Adolescent (PNSA). Therefore, the research team is not obliged to seek approval from the university of Kinshasa. 

Answer: The approved research report and datasets will be posted on the WHO website (www.who.int) and Population and Health Research Institute (PHERI) website (www.pheri.org) at the end of the study. Interested researchers will apply for authorisation access. 

Answer: This is a protocol paper. We do not have any data collected yet. As we indicated, at the end of this study, the approved report and datasets will be posted on the WHO website (www.who.int) and Population and Health Research Institute (PHERI) website (www.pheri.org).

Answer: Data will be property of WHO. We will provide a request form to access the data. We will update your Data Availability statement to reflect the information you provide in your cover letter.

Answer: We have provided ORCID iD for the corresponding author in Editorial Manager.

https://orcid.org/0000-0003-3709-1121

6. Your ethics statement must appear in the Methods section of your manuscript. If your ethics statement is written in any section besides the Methods, please move it to the Methods section and delete it from any other section. Please also ensure that your ethics statement is included in your manuscript, as the ethics section of your online submission will not be published alongside your manuscript.

Answer: Thank you for the comment. We have moved the ethics statement into the methods section. 

7.We note that [Figure(s) 2] in your submission contain [map/satellite] images which may be copyrighted. All PLOS content is published under the Creative Commons Attribution License (CC BY 4.0), which means that the manuscript, images, and Supporting Information files will be freely available online, and any third party is permitted to access, download, copy, distribute, and use these materials in any way, even commercially, with proper attribution. For these reasons, we cannot publish previously copyrighted maps or satellite images created using proprietary data, such as Google software (Google Maps, Street View, and Earth). For more information, see our copyright guidelines: http://journals.plos.org/plosone/s/licenses-and-copyright.

• You may seek permission from the original copyright holder of Figure(s) [2] to publish the content specifically under the CC BY 4.0 license. We recommend that you contact the original copyright holder with the Content Permission Form (http://journals.plos.org/plosone/s/file?id=7c09/content-permission-form.pdf) and the following text:

o “I request permission for the open-access journal PLOS ONE to publish XXX under the Creative Commons Attribution License (CCAL) CC BY 4.0 (http://creativecommons.org/licenses/by/4.0/). Please be aware that this license allows unrestricted use and distribution, even commercially, by third parties. Please reply and provide explicit written permission to publish XXX under a CC BY license and complete the attached form.”

o Please upload the completed Content Permission Form or other proof of granted permissions as an "Other" file with your submission.

o In the figure caption of the copyrighted figure, please include the following text: “Reprinted from [ref] under a CC BY license, with permission from [name of publisher], original copyright [original copyright year].”

• If you are unable to obtain permission from the original copyright holder to publish these figures under the CC BY 4.0 license or if the copyright holder’s requirements are incompatible with the CC BY 4.0 license, please either i) remove the figure or ii) supply a replacement figure that complies with the CC BY 4.0 license. Please check copyright information on all replacement figures and update the figure caption with source information. If applicable, please specify in the figure caption text when a figure is similar but not identical to the original image and is therefore for illustrative purposes only. The following resources for replacing copyrighted map figures may be helpful:

o USGS National Map Viewer (public domain): http://viewer.nationalmap.gov/viewer/

o The Gateway to Astronaut Photography of Earth (public domain): http://eol.jsc.nasa.gov/sseop/clickmap/

o Maps at the CIA (public domain): https://www.cia.gov/library/publications/the-world-factbook/index.html and https://www.cia.gov/library/publications/cia-maps-publications/index.html

o NASA Earth Observatory (public domain): http://earthobservatory.nasa.gov/

o Landsat: http://landsat.visibleearth.nasa.gov/

o USGS EROS (Earth Resources Observatory and Science (EROS) Center) (public domain): http://eros.usgs.gov/#

o Natural Earth (public domain): http://www.naturalearthdata.com/

o

Answer: Thank you for the comment. We have removed Figure 2. 

Reviewer #1: Dear authors,

The protocol for the situation analysis of SRHS for displaces population of Kasaï is an interesting topic that deserves full assessment to improve such health care delivery for displaced women. From the description this project is very ambitious and a huge amount of work will be required to pull this off, I’m assuming that the authors have sufficient time, human and financial resources to fulfil all the work.

The manuscript needs substantial revision for the English language, the manuscript should be edited by an English native speaker.

My concern in general is that this situation analysis concerns SHR health care delivery. However, from the protocol it was not very clear if the situation analysis of health care facilities that will be visited, if they were specifically set-up in these camps to deliver health care for the displaces population? Or if staff has received specific training for the care of post-conflict, displaced individuals? From reading the manuscript, the feeling is that it is a general SHR assessment of SHR rather than SHR assessment for displaced people. It seems that the health care facilities that will be visited and audited are the ones that are traditionally there to provide healthcare to the resident population. Then the outcome could be biased because of course these type of healthcare facilities will not be prepared to deliver post-traumatic healthcare.

Another concern is the focus of the study, data on SHR delivery will be collected from a large variety of sources, which is an advantage for triangulation, however the questionnaires of Qualitative research are very exhaustive collecting insights on SHR delivery for men, women, adolescent boys and girls. From the introduction I understood that the target study population was displaced women, while the Qualitative study is trying to gather insight on SHR delivery in general, this related back to my first concern. Do you want to know about SHR delivery in general? Or specifically for the displaced female population?

It is not explained in the text why you are sampling women and young girls from 12-49 years of age for the women questionnaire while the WHO definition of women of reproductive age is 15-49 years of age.

Lastly, my major concern is about the sampling of households and women for the community-based survey. In the methods you describe that you will select from an updated list of villages/camps. Already in the introduction it is not well described if the displaced are living in villages (permanent housing) or in camps (temporary housing). The description of the context with a bit more detail will help to understand the methodology used. Later on in the expected challenges section you describe that the location and the ability to identify and recruit the target populations could be a challenge. The mitigation plan is to involve leaders and NGO’s, but my concern is how will you be able to apply the random sampling strategy if this housing list, village list and camp list are not available?

Answer: Thank you for your comment. As we stated in the protocol, displaced people are living in specific neighbourhood of some villages. During the crisis, people were moving to neighboring villages / territories and at one point, the crisis became widespread. Displaced people often lived among the population. However, they built their houses in well located areas in the villages. As a result, the entire population used the same health structures, which were overwhelmed because initially, these facilities were not prepared for humanitarian emergency. Non-Governmental Organizations and UN institutions (UNFPA, WHO, UNICEF, OCHA, etc.) trained nursing staff from the village health center, as well as some nursing staff among the displaced people on the Minimum Emergency System and other care. Therefore, their identification is not a problem. 

More specific comments will be uploaded in a separate file.

1. The protocol for the situation analysis of SRHS for displaces population of Kasaï is an interesting topic that deserves full assessment to improve such health care delivery for displaced women. From the description this project is very ambitious and a huge amount of work will be required to pull this off, I’m assuming that the authors have sufficient time, human and financial resources to fulfil all the work.

Answer: Thank you! We appreciate your positive feedback. This work is funded by the Department of Sexual and Reproductive Health and Research (SRH), including the UNDP/UNFPA/UNICEF/WHO/World/Bank Special program of research, development and research training in human reproduction (HRP), from the generous support of the Ministry of Foreign Affairs of the Netherlands. The funders had no role in study design, data collection and analysis, decision to publish, or preparation of the manuscript.

2. The manuscript needs substantial revision for the English language, the manuscript should be edited by an English native speaker. 

Answer: Thank you for the comment which we have addressed.

3. My concern in general is that this situation analysis concerns SHR health care delivery. However, from the protocol it was not very clear if the situation analysis of health care facilities that will be visited, if they were specifically set-up in these camps to deliver health care for the displaces population? Or if staff has received specific training for the care of post-conflict, displaced individuals? From reading the manuscript, the feeling is that it is a general SHR assessment of SHR rather than SHR assessment for displaced people. It seems that the health care facilities that will be visited and audited are the ones that are traditionally there to provide healthcare to the resident population. Then the outcome could be biased because of course these type of healthcare facilities will not be prepared to deliver post-traumatic healthcare. 

Answer: Thank you for your comment. As we indicated in the protocol, displaced people are living in specific neighbourhood of some villages. During the crisis, people were moving to neighboring villages / territories and at one point, the crisis became widespread. Displaced people often lived among the population. However, they built their houses in well-known areas in the villages. As a result, the entire population including the displaced population used the same health system and facilities, which were overwhelmed because initially, these facilities were not prepared for humanitarian emergency. Non-Governmental Organizations and UN institutions (UNFPA, WHO, UNICEF, OCHA, etc.) trained nursing staff from the village health center, as well as some nursing staff among the displaced people on the Minimum Emergency System and other care. Therefore, their identification is not a problem. 

4. Another concern is the focus of the study, data on SHR delivery will be collected from a large variety of sources, which is an advantage for triangulation, however the questionnaires of Qualitative research are very exhaustive collecting insights on SHR delivery for men, women, adolescent boys and girls. From the introduction I understood that the target study population was displaced women, while the Qualitative study is trying to gather insight on SHR delivery in general, this related back to my first concern. Do you want to know about SHR delivery in general? Or specifically for the displaced female population? 

Answer: This study targets displaced adolescent girls and women (aged 12-49). However, to better understand the problem and design effective as well as responsive programs (interventions), we aim to consider the perceptions and point of views of men using FGDs. Such approach will consider gender norms in promoting SRHR in the humanitarian context. As we mentioned in the background, child marriage (before 15) and Gender-based violence are common in that context. Furthermore, adolescent girls aged less than 15 might face specific SRH challenges such as symptoms during menstruation, menstruation hygiene, transactional sexual intercourse, etc. 

5. It is not explained in the text why you are sampling women and young girls from 12-49 years of age for the women questionnaire while the WHO definition of women of reproductive age is 15-49 years of age. 

Answer: Studies of sexual and reproductive health (SRH) in the Democratic Republic of Congo (DRC) have often included adolescent girls under the age of 15 because of high prevalence of child marriage and early onset of childbearing, especially in the humanitarian context. According to the 2013 Demographic and Health Survey (DHS), about 16% of surveyed women got married by age 14 while the prevalence of early child marriage (marriage by 15) was estimated at 30%. We would like to capture the perspectives of adolescent girls, their needs as well as their concerns, to which data is currently lacking in DRC among this displaced population.

6. Lastly, my major concern is about the sampling of households and women for the community-based survey. In the methods you describe that you will select from an updated list of villages/camps. Already in the introduction it is not well described if the displaced are living in villages (permanent housing) or in camps (temporary housing). The description of the context with a bit more detail will help to understand the methodology used. Later on in the expected challenges section you describe that the location and the ability to identify and recruit the target populations could be a challenge. The mitigation plan is to involve leaders and NGO’s, but my concern is how will you be able to apply the random sampling strategy if this housing list, village list and camp list are not available? 

Answer: Thank you for your comment. Though displaced people are living in the villages, community leaders and humanitarian workers know their specific locations. We will seek their collaboration to obtain the needed list of households and villages (specifically, the chiefs of villages usually have access to this information). Based on that list, we will draw the needed random sample. This was explained in the manuscript.

Specific comments on the manuscript

Abbreviations should be written in full the first time that it is used, not after (Also in the abstract)

7. Introduction: “ The region had the highest fertility (8 children per women versus the national average of 6.6 children per woman); significantly low modern contraceptive prevalence (6%) and relatively low unmet needs for FP.” This part of the sentence doesn’t make sense, the proportion relates to unmet needs? Or the needs that were met? 

Answer: Thank you for your comment. We have deleted the last part of the sentence as you suggested. However, low contraceptive prevalence is not necessary associated with high unmet needs, based on the perceptions of women in Kasai. In general, it has been cited that even women with perceived ideal number of children do not wish to prevent future pregnancies, as children, from a cultural perspective, as gift from God. Unmet needs appear only when additional pregnancy represent a threat (for the health of the mother or child heath) or as socioeconomic threat. 

8. Study design: the first paragraph is quite confusing while reading. combining community-based survey with standard assessment does not reflect mixed method. I suggest to break this into multiple phrases. And actually, your study design is 3 surveys: 1) community based survey (QUANT); 2) health facility assessment (QUANT) and 3) Key informant interviews and FGD (QUAL). Bringing this structure to the manuscript would much more comprehensive for the reader. 

Answer: Thank you for your comment. However, our first sentence stated that: “This research will consist of a mixed-methods study design, combining quantitative and qualitative approaches”. We did not say that mixed method means combination of community-based survey with standard assessment. For more clarity we have revised the paragraph and we outlined that the quantitative components will include a community-based survey with adolescent girls and women as well health facilities’ assessment, while the qualitative component will include a series of KIIs and FGDs. The purpose of these two approached is to triangulate the findings from both arms which will provide a comprehensive perspective on the prevailing SRH issues, needs, and health system response among the displaced population in the Kasai.

9. Participants: “the study will target different population” should be ‘target different study subjects’

You should describe better which study population will be targeted for which survey or questionnaire.

Answer: Thank you for your comment. We have revised the paragraph accordingly. 

10.Data sources: 

- Will you recruit 500 households, or 500 women and young girls? The way you describe, more than 1 women can live in a household. Therefore, the number of households in unknown, and you will recruit until you have reached the minimal sample size = 500 women and young girls.

Answer: Thank you for your question. We will select 500 households in each province. All adolescent girls and women aged 12-49 will be eligible for community-based household interviews in the three provinces. 

- “In addition, to the household and women questionnaire, a health facility assessment will be

administered. The health facility assessment will employ a health facility quantitative survey coupled with a facility audit.” This is very confusing as a transition sentence, it is as if you are going to administer this assessment to the same study population. You can separate these two surveys. 

- Answer: Thank you for your comment. We have deleted the first sentence : “In addition, to the household and women questionnaire, a health facility assessment will be administered”.

- “In each selected health facility, an audit, a provider’s interview, a non-participatory observation and client exit interview will be conducted.” The two activities in bolt, I have serious concern about this, due to confidentiality reasons. Especially since it relates to SRH, this might be very uncomfortable for the study subjects. You don’t mention it later on in the protocol, is this maintained or not? Personally, I don’t agree for these methods for SRH. 

Answer: Thank you for your comment. We have taken your concerns into consideration and accordingly removed non-participatory observation and client exit interview from the protocol.

- “A total sample of 60 providers in each site will be interviewed”; this is confusing as later on you say you will recruit 4 health care providers per site? 

Answer: Thank you for your comment. We have clarified the sentence: “A total sample of 60 providers in each site (4 providers X 15 health facilities) will be interviewed”. 

- The questionnaire for provider interview was not provided, you say you will assess “experience and perceptions” I’m assuming this is a quantitative questionnaire, therefore perception cannot be assessed, but rather attitudes and practices, like a (Knowledge, attitudes and practices) KAP design.

Answer: Thank you for your comment. We have revised the sentence and use attitudes as you suggested. 

- Audit of the facility: read my concern on assessment of facilities in general? Or specifically targeted to displaced population?

Answer: Thank you for your comment. There are no specific facilities for displaced people. The study will assess health facilities located in the village where displaced people are living. The displaced people are using some of the same health facilities as those of the local population, which are readily identifiable in each village. We will not select health facilities in villages that the displaced people are not using.

- First time you mention FGD, it is not clear if men and women will be interviewed separately, you explain later on in the text that you will, but you should be clear from the start that this is a delicate subject, so women and men should not be mixed in the FGD. 

Answer: Thank you for your comment. The FGDs will be conducted separately fo both men and women. Also, older men and women FGDs will not be mixed with younger adolescent girls and boys.

- You should mention that FDGs are conducted until saturation of information is reached. It is highly doubtful that with the sample size of 12FGD you will reach this saturation. Especially when you split up as you describe. 4 FGD per province, 2 in men and 2 in women, and then from the two, you split into 1 in adults and 1 in youth. This means you only have 1 FGD per age group per sex. Re-consider the minimum sample size if you want to maintain this methodology. 

Answer: Thank you for your comment. We have doubled the sample size for FGDs. In total we will conduct 24 FGDs (8 per province). Specifically, we will conduct two FGDs per stratum (Female <25; Female 25&+; Male <25; Male 25&+) per province. This is consistent with Hennink (2019) who suggested that two groups per stratum provided a more comprehensive understanding of issues, while more groups per stratum provided little additional benefit, particularly in mixed studies. 

11. Ethical considerations

You will take informed verbal consent for everybody?

I agree for the qualitative research, you can maintain verbal consent. 

Answer: Thank you

I disagree for the quantitative data. Especially in the health care facilities, providers are nurses and doctors, they should be able to sign, written informed consent can be asked here.

Answer: Thank you for the suggestion which we will implement accordingly.

For the household survey written informed consent would be preferable, with finger print if respondents cannot read and write, however, verbal consent can justify. This justification should be mentioned. 

Answer: Thank you for your comment. We will keep verbal consent because a potential high number of the displaced women in Kasai are illiterate and cannot write. It should be noted that this is in accordance with the Demographic and Health Survey (DHS) and the Multiple Indicators Cluster Survey (MICS). These two surveys also used verbal consent.

---

## [Decision Letter · Decision Letter 1]

27 Oct 2020

Situation Analysis for Delivering Integrated Comprehensive Sexual and Reproductive Health Services for displaced population of Kasaï, Democratic Republic of Congo: Protocol for a mixed method study

PONE-D-20-10507R1

Dear Dr. Emina,

We’re pleased to inform you that your manuscript has been judged scientifically suitable for publication and will be formally accepted for publication once it meets all outstanding technical requirements.

Kind regards,

Philip Anglewicz, PhD

Academic Editor

PLOS ONE

Additional Editor Comments (optional):

Reviewers' comments:

Reviewer's Responses to Questions

**Comments to the Author**

1. Does the manuscript provide a valid rationale for the proposed study, with clearly identified and justified research questions?

Reviewer #1: Yes

2. Is the protocol technically sound and planned in a manner that will lead to a meaningful outcome and allow testing the stated hypotheses?

Reviewer #1: Yes

3. Is the methodology feasible and described in sufficient detail to allow the work to be replicable?

Reviewer #1: Yes

4. Have the authors described where all data underlying the findings will be made available when the study is complete?

Reviewer #1: Yes

5. Is the manuscript presented in an intelligible fashion and written in standard English?

Reviewer #1: Yes

6. Review Comments to the Author

You may also provide optional suggestions and comments to authors that they might find helpful in planning their study.

Reviewer #1: Congratulations,

the manuscript has improved immensly

Correct some minor mistakes, like mentionning the abbreviations in full the first time in the introduction

Under the title Quantitative survey, no need to mention quantitative again in Community based-survey, and Health Facility surveys

And for the FGD: the proposition is good, however the FGD method is also usually applied until saturation of information, therefore I suggest to mention this as well in the manuscript that if saturation is not reached after 2 FGD per age group.

7. PLOS authors have the option to publish the peer review history of their article (what does this mean?). If published, this will include your full peer review and any attached files.

Reviewer #1: No

---

## [Editor Report · Acceptance letter]

17 Nov 2020

PONE-D-20-10507R1 

Situation Analysis for Delivering Integrated Comprehensive Sexual and Reproductive Health Services for displaced population of Kasaï, Democratic Republic of Congo: Protocol for a mixed method study 

Dear Dr. Emina:

I'm pleased to inform you that your manuscript has been deemed suitable for publication in PLOS ONE. Congratulations! Your manuscript is now with our production department. 

Kind regards, 

on behalf of

Associate Professor Philip Anglewicz 

Academic Editor

PLOS ONE